# Holothurin A Inhibits RUNX1-Enhanced EMT in Metastasis Prostate Cancer via the Akt/JNK and P38 MAPK Signaling Pathway

**DOI:** 10.3390/md21060345

**Published:** 2023-06-03

**Authors:** Sirorat Janta, Kanta Pranweerapaiboon, Pornpun Vivithanaporn, Anuchit Plubrukarn, Arthit Chairoungdua, Prachayaporn Prasertsuksri, Somjai Apisawetakan, Kulathida Chaithirayanon

**Affiliations:** 1Department of Anatomy, Faculty of Science, Mahidol University, Bangkok 10400, Thailand; sirorat.jat@student.mahidol.ac.th (S.J.); kantapra@tu.ac.th (K.P.); prachayaporn.prs@student.mahidol.ac.th (P.P.); 2Chulabhorn International College of Medicine, Thammasat University, Pathumthani 12120, Thailand; 3Chakri Naruebodindra Medical Institute, Faculty of Medicine, Ramathibodi Hospital, Mahidol University, Bangkok 10540, Thailand; pornpun.viv@mahidol.edu; 4Department of Pharmacognosy and Pharmaceutical Botany, Faculty of Pharmaceutical Sciences, Prince of Songkla University, Songkhla 09112, Thailand; anuchit.pl@psu.ac.th; 5Department of Physiology, Faculty of Science, Mahidol University, Bangkok 10400, Thailand; arthit.chi@mahidol.ac.th; 6Department of Anatomy, Faculty of Medicine, Srinakharinwirot University, Wattana, Bangkok 10110, Thailand; somjaia@g.swu.ac.th

**Keywords:** RUNX1, holothurin A, EMT, metastasis, prostate cancer

## Abstract

Due to the challenge of prostate cancer (PCa) management, there has been a surge in efforts to identify more safe and effective compounds that can modulate the epithelial–mesenchymal transition (EMT) for driving metastasis. Holothurin A (HA), a triterpenoid saponin isolated from *Holothuria scabra*, has now been characterized for its diverse biological activities. However, the mechanisms of HA in EMT-driven metastasis of human PCa cell lines has not yet been investigated. Moreover, runt-related transcription factor 1 (RUNX1) acts as an oncogene in prostate cancer, but little is known about its role in the EMT. Thus, the purpose of this study was to determine how RUNX1 influences EMT-mediated metastasis, as well as the potential effect of HA on EMT-mediated metastasis in endogenous and exogenous RUNX1 expressions of PCa cell lines. The results demonstrated that RUNX1 overexpression could promote the EMT phenotype with increased EMT markers, consequently driving metastatic migration and invasion in PC3 cell line through the activation of Akt/MAPK signaling pathways. Intriguingly, HA treatment could antagonize the EMT program in endogenous and exogenous RUNX1-expressing PCa cell lines. A decreasing metastasis of both HA-treated cell lines was evidenced through a downregulation of MMP2 and MMP9 via the Akt/P38/JNK-MAPK signaling pathway. Overall, our approach first demonstrated that RUNX1 enhanced EMT-driven prostate cancer metastasis and that HA was capable of inhibiting the EMT and metastatic processes and should probably be considered as a candidate for metastasis PCa treatment.

## 1. Introduction

Prostate cancer (PCa) is considered to be the most prevalent malignancies in men and the majority of PCa-related deaths are generally attributed to metastasis [1,2]. Metastasis represents a complex multistep process by which cancer cells leave the primary occurrence, invade adjacent tissues, travel through the blood circulation, and eventually establish metastatic colonies at distant organs. The epithelial-to-mesenchymal transition (EMT) is a critical mechanism by which epithelial cells reprogram and transform into a spindle-shaped mesenchymal phenotype and is involved in the onset of metastasis [3]. During the EMT at the molecular level, tumor cells switch their epithelial markers, including epithelial cadherin (E-cadherin), zona occludens-1 (ZO-1), and claudin into mesenchymal markers, such as neural cadherin (N-cadherin) fibronectin and vimentin. Particularly, E-cadherin, an important cell–cell adhesion molecule in intercellular crosstalk, is a key master for the EMT initiation that is inhibited by EMT-related transcription factors (TFs), including twist-related protein1 (TWIST1), SnaiI1 (Snail), SnaiI2 (Slug), and zinc finger E-box-binding homeobox1 and 2 (ZEB1 and ZEB2) [4]. Furthermore, the upregulation of these EMT-related transcription factors could increase matrix metalloproteinases (MMPs), leading to tumor cell metastasis [5]. 

Presently, runt-related transcription factors (RUNX1, RUNX2, and RUNX3) are a group of metazoan transcription factors that act as master regulators of a variety of cellular processes and development. The RUNX genes have received increased attention recently due to their role in tumorigenesis [6]. Intriguingly, RUNX1 acts as a tumor suppressor or as a dominant oncogene of the EMT process based on different types of cancer [6,7,8,9,10,11]. For example, RUNX1 sustains a normal epithelial phenotype and suppresses E-cadherin expression, leading to inhibiting the EMT in breast cancer [12]. Conversely, RUNX1 expression induced by TGF-β was high in colorectal cancer tissues (CRC) and its upregulation in vitro and in vivo could expedite CRC cell metastasis by governing the EMT [9,10]. Considering the PCa, all we know is that RUNX1 and RUNX2, but not RUNX3, are highly expressed in PCa tissues, the primary prostatic carcinoma cell line, and the LNCaP and PC3 PCa cell lines [13]. Moreover, both RUNX1 and RUNX2 can promote MMP2/MMP9 expression in PCa metastasis [11]. Nonetheless, how RUNX1 promotes the EMT for driving PCa metastasis is not well understood. As a result, elucidating their regulatory mechanisms may add a new dimension to our understanding of PCa progression.

Nowadays, marine natural products have become an alternative treatment option for cancer treatment [14]. Sea cucumbers have long been used in Asian countries as food and traditional medicine. Holothurin A (HA) is a triterpene glycoside widely distributed along with sea cucumbers belonging to the family Holothuriidae. It firstly was isolated from *Actynopyga agassizi* [15], but its full structure was elucidated by Kitagawa et al. on the sample isolated from *Holothuria leucospilota* [16] (Figure 1) and has been characterized for the presence of various biological activities [17,18]. Of note, our previous study addressed that HA had a substantial cytotoxic effect on the androgen-positive LNCaP cell line in accordance with the reduction of the androgen receptor expression, resulting in the downregulation of the prostate-specific antigen [19]. However, the anticancer effect of HA against PCa remains unknown, particularly in terms of the EMT for the metastasis cascade. Thus, the purpose of this study was to elucidate the inhibitory effects of HA on EMT-enhanced metastasis through the endogenous and exogenous RUNX1 overexpression of human PCa cell lines with the related molecular mechanism. As a result, this could bring the identification of a novel therapeutic candidate for the treatment of metastatic prostate cancer.

## 2. Results

### 2.1. HA Promotes Cytotoxicity in Human PCa Cell Lines

To determine whether HA affects the viability of human PCa cells, an MTT assay was used in this experiment. Here, PC3 cells were firstly treated with various concentrations of HA from 0.3125 to 20 µM for 24 and 48 h. It was found that HA could greatly inhibit the viability on PC3 cells when compared with the control. The half-maximal inhibitory concentration (IC_50_) values at 24 and 48 h were 1.68 and 1.56 µM for PC3 (Figure 2A). As stated in the IC_50_ values of HA for PC3 cells, consequently, different concentrations at 0.25, 0.5, and 1 µM HA were chosen as appropriate dosages for a further comparative morphological investigation. The marked morphological changes in HA-treated PC3 showed a shrinkage of cells and a lack of intercellular contact compared with the nontreated group (Figure 2B).

In our previous study [19], HA exerted a cytotoxicity in LNCaP cells, reporting an IC_50_ at 24 and 48 h of 1.98 and 1.59 µM, respectively. The same concentration at 0.25–1 µM in LNCaP cells that were also affected by HA showed morphological changes in the cells and nuclei (Figure 2C). These data indicated that HA augmented the growth-inhibitory effect on prostate cancer cell lines (PC3 and LNCaP) in a concentration-dependent manner. 

In addition, we also compared the efficacy and toxicity of docetaxel with HA in PC3 and LNCaP cells analyzed by an MTT assay. The usage of docetaxel from 0.001 to 5 µM revealed that IC_50_ values at 24 and 48 h were 2.01 and 0.45 µM for PC3, and 2.20 µM and 0.001 µM for LNCaP cells (Appendix A). These suggested that at 24 h, HA suppressed the viabilities of both PC3 and LNCaP cells more substantially than docetaxel, an active drug in the treatment of PCa.

### 2.2. RUNX1 Promotes EMT Process Driving for Metastasis in PCa Cell Line

Ectopic upregulated RUNX1 has been pinpointed as contributing to the carcinogenesis of various cancer cells. In a PCa cell line, PC3 cells are well known for having highly aggressive and metastatic phenotypes, being independent of androgen, and belonging to castration-resistant prostate cancer. To test this notion, the biological activity of overexpressing RUNX1 in PC3 cells was established. At twenty-four hours post-transfection of RUNX1, a qRT-PCR and Western blot revealed a massive increase in RUNX1 mRNA (Figure 3A) and protein expressions (Figure 3B) in RUNX1-overexpressing PC3 cells when compared to untransfected control cells. As expected, the analysis of EMT markers using qRT-PCR demonstrated that E-cadherin was significantly downregulated in RUNX1-overexpressing PC3 cells relative to control (Figure 4A). On the contrary, the upregulation of vimentin, MMP2, and MMP9 were enhanced in PC3 cells with the overexpression of RUNX1 (Figure 4A). Consistent with these results, Western blotting revealed a decrease in the expression of E-cadherin, whereas the expression levels of RUNX1, vimentin, and EMT-related transcription factors, including twist1, slug, and snail, were upregulated in ectopic RUNX1-transfected PC3 cells (Figure 4A). Collectively, these findings verified that RUNX1 could enhance EMT markers in PC3 at the transcription and protein levels.

### 2.3. HA Suppresses mRNA and Protein Expressions of EMT-Related Transcription Factors 

Because RUNX1 overexpression led to an increase in EMT markers and MMPs expressions, we first validated the potential effect of HA on the regulation of the ectopic RUNX1-enhanced EMT as well as endogenous RUNX1 in PC3 and LNCaP cell lines in our recent study. Markedly, the involvement of HA at 0.5 µM could dramatically decrease the relative mRNA expression levels of RUNX1, vimentin, MMP2, and MMP9; conversely, it increased the gene expression of E-cadherin in the RUNX1 overexpression of PC3 cells (Figure 4A). Consistent with this approach, Western blotting revealed that an HA treatment with the indicated concentrations (0.25–1 µM) in the ectopic expression of RUNX1 in PC3 cells decreased protein expressions of the mesenchymal marker vimentin, and EMT-related transcription factors, including twist1, slug, and snail, but not the epithelial marker E-cadherin (Figure 4A).

In addition, the HA treatment in both PC3 and LNCaP cell lines exerted a significant upregulation of the E-cadherin gene. In contrast, the expression of mesenchymal marker genes such as N-cadherin, vimentin, twist1, slug, and snail were downregulated in PC3 and LNCaP cells compared with the control group (Figure 4B,C). A similar Western blot pattern was also observed in the PC3 and LNCaP cells, where the administration of each concentration of HA resulted in E-cadherin upregulation, while RUNX1, vimentin, twist1, slug, and snail expressions were decreased (Figure 4B,C). Obviously, as a result of the downregulation of RUNX1 expression, it might be implied that HA could impair RUNX1 in both PCa cell lines as well as with RUNX1 overexpression. Altogether, these conclusions provided strong evidence that the HA treatment impeded the induction of EMT-mediated metastasis in the endogenous and ectopic RUNX1 expression of PCa cell lines. 

### 2.4. HA Treatment Suppresses Migration and Invasion of PCa Cell Lines as Well as RUNX1 Overexpression

To confirm the influence of HA on the enhancement of metastasis via ectopic RUNX1 expression, the migration and invasion of RUNX1-overexpressing PC3 cells were manipulated using transwell migration and Matrigel invasion assays. Results revealed that overexpression of RUNX1 in PC3 cells triggers an increase in migration and invasion abilities (Figure 5A). Evidently, RUNX1-enhanced cell migration and invasion were both suppressed after being treated with 0.5 and 1 µM HA in a dose-dependent manner, as compared with the control and docetaxel-treated groups (Figure 5A).

In parallel with these results, a treatment of HA at the same concentration could considerably reduce the migratory and invasive capacity of PC3 and LNCaP cells, as shown in the decreasing percentage of migrated and invaded cells compared with the untreated group (Figure 5B,C). Therefore, these results evidenced markedly that HA virtually decreased the migration and invasion of both PCa cell lines as well as RUNX1 overexpression.

### 2.5. HA Diminishes the Gelatinolytic Activities of MMPs in Pca Cell Lines as Well as RUNX1 Overexpression

Much evidence has been reported about several matrix metalloproteinases (MMPs) of cancer, such as MMP2 and MMP9, driving cancer invasion by degrading extracellular matrices. Therefore, a Western blot analysis and gelatin zymography were performed in both PCa cell lines as well as in the RUNX1 overexpression in PC3 cells. After transfection, the expressions of MMP2 and MMP9 were increased with the RUNX1 overexpression in PC3 cells when compared to the control group (Figure 6A). On the contrary, the HA treatment, at a concentration of 1 µM, could greatly downregulate the expressions of MMP2 and MMP9 compared with the RUNX1-transfected control and docetaxel-treated groups (Figure 6A). A similar result was also found where the MMP9 gelatinolytic activity was largely attenuated by the HA treatment, in particular at 1 µM, compared with the control groups (Figure 6A).

Similarly, a Western blot detected that the administration of HA at concentrations higher than 0.5 µM on PC3 and LNCaP cells reduced the expression of MMP2 and MMP9 in a concentration-dependent manner compared with the control group (Figure 6B,C). Moreover, MMP2 and MMP9 mRNA expressions in the PCa cell lines were essentially decreased following 0.5 µM of HA treatment (Figure 6B,C). The HA treatment at 1 µM concurrently attenuated the MMP9 gelatinase activity of both PCa cell lines as compared with the control (Figure 6B,C). Taken together, HA has an efficacy to suppress the migratory and invasion potential of prostate cancer PC3 and LNCaP cells as well as overexpressing RUNX1 in PC3 cells via the downregulation of MMP2 and MMP9. 

### 2.6. HA Inhibits EMT-Mediated Metastasis via Inhibiting Akt, JNK, and P38 MAPK Signaling Pathways

Considering the critical signaling mechanism that regulates RUNX for the EMT for driving metastasis, the MAPK signaling pathway was one of the most intriguing possibilities studied. Remarkably, we noticed a significant increase in p-Akt in the overexpressing RUNX1 of the PC3 cell line. As a consequence of the activation of the p-Akt upstream, the MAPK signaling cascade, including p-ERK, p-JNK, and p-P38 were obviously increased as compared to the control group (Figure 7A). Next, we further evaluated HA on the MAPK signaling pathway in endogenous and ectopic RUNX1. The present findings evidenced that HA truly showed a consistent longitudinal downregulation of the phosphorylation levels of Akt, JNK, and P38 in RUNX1-overexpressed PC3 cells (Figure 6A) as well as in PC3 and LNCaP cells (Figure 7B,C) as compared to the control group. Yet, there was no obvious change in the phosphorylation level of ERK in response to the HA treatment in these cells (Figure 7A,C). 

Hence, the results proved that HA inhibited the activation of EMT-mediated metastasis through the Akt, JNK, and P38 MAPK signaling pathways in PCa cell lines and even in ectopic RUNX1 overexpression. 

## 3. Discussion

The primary activation of the EMT has emerged as a key event for cancer metastasis, which usually develops into a poor prognosis for patients [2,3,20]. Thus, inhibiting EMT-driven cancer metastasis would be expected to improve overall survival in patients. Herein, this report is the first in-depth analysis showing that HA can suppress the RUNX1-enhanced EMT for metastasis in both PCa cell lines via the Akt, JNK, and P38 MAPK signaling pathways. 

Notably, the EMT is a critical process in which epithelial cells lose their apical–basal polarity, become elongated, and increase their motility, resulting in an increased capacity for cellular migration and invasion [21,22]. Various transcriptional regulators have served as oncogenes and the tumor suppressor genes of EMT have been addressed and targeted for cancer therapy. Interestingly, RUNX1 accounts for the role of oncogene in the EMT and metastasis of several types of cancer [23]. For example, RUNX1 activation in colorectal cancer correlated with various possible mechanisms, including the Wnt/β-catenin signaling pathway [9] and TGF-β induction through the Smad-dependent pathway and Smad-independent pathway [10]. Moreover, RUNX1 could promote partial EMT marker genes via increasing the transcription of the PI3K subunit p110δ after the TGF-β induction in renal tubular epithelial cells [24]. That was confirmed by the PI3K inhibitor (LY294002) and the p110δ inhibitor (CAL-101), which inhibited the enhancement of N-cadherin. Consistent with our findings, the overexpression of RUNX1 did enhance the EMT and metastasis in PC3 cells as compared with the control group. We noticed that the significant increased expression of p-Akt with its signaling cascade was most likely due to the elevated expression of RUNX1, thereby facilitating the EMT process. A recent report demonstrated that Akt was required for the regulation of snail and slug’s expression, two critical transcription factors of EMT, via various signaling cascades. For instance, the activation of Akt can phosphorylate glycogen synthase kinase 3 (GSK-3), thereby promoting GSK-3 degradation and enhancing the snail and slug stability [25]. Additionally, Akt can promote the nuclear factor-B subunit p65, resulting in an increase in snail transcription [26]. Meanwhile, our findings indicated that RUNX1 could promote the EMT and metastasis via the activation of the Akt, JNK, and P38 MAPK signaling pathways, whereas ERK expression remained unchanged. Likewise, numerous upstream signaling cascades, including the MAPK pathway, influence the activation of RUNXs in human cancers [27]. Recent reports indicated that the JNK/P38 MAPK signaling pathway was critical for the tumor EMT and metastasis that regulates the expression of EMT-associated proteins, such as E-cadherin and vimentin [28,29,30]. Several studies have demonstrated JNK and P38’s ability to promote PCa metastasis by the induction of MMP2 and MMP9, as well as u-PA [31,32,33,34]. It was also found that the inhibition of JNK could suppress EMT-related signaling targets in DU145 PCa cells [31]. On the contrary, an extracellular signal-regulated kinase (ERK) is involved with proliferation and antiapoptosis in prostate cancer [31]. Moreover, our current results were consistent with the previous findings from Sangpairoj et al. [7], in which RUNX1 knockdown interfered with its nuclear translocation and could suppress migration and invasion as well as angiogenesis by downregulating MMPs and VEGFA expression through the P38 MAPK signaling pathway. Thus, our collective data suggest that RUNX1 acts as an oncogene in PCa by regulating EMT markers, such as E-cadherin, vimentin, slug, snail, and twist1, as well as MMP2 and MMP9, via the activation of the Akt/JNK and P38 MAPK signaling pathways.

According to our data, a correlation between RUNX1 expression and MMP2 and MMP9 protein expression levels with its gelatinolytic activity could lead to increase in the migration and invasive ability of the PC3-overexpressing cell line. These are consistent with the previous evidence that the primary function of the RUNXs in PCa is to regulate MMPs [35,36,37]. Cancer cells must breach the ECM in order to exit the primary tumor site, which is facilitated by a variety of ECM-degrading enzymes, including MMPs as endopeptidases [38]. In particular, MMP9 has been identified as a crucial metastatic factor in PCa [36]. Prior studies have addressed that a 250 bp promoter site on the MMP9 gene appears to be a specific target of RUNX1 as well as RUNX2 [36]. Moreover, the overexpression of miR-141, which targets RUNX1, inhibits PCa cell proliferation and increases proapoptotic factors such as FOXO1 and p21. Both MMP2 and MMP9 levels decrease in response to miR-141 upregulation, resulting in the suppression of cell migration and invasion. However, these effects are reversed in the presence of RUNX1 [11]. Together, these data suggest a mechanistic relationship between RUNX1 and MMP2 and MMP9 for tumor prostate cancer cell metastasis. 

The high mortality rate associated with PCa is primarily due to therapeutic resistance and rapid metastasis [1,2,20]. Thus, our recent study concentrated on the discovery of novel compound derived from natural products that are capable of overcoming PCa intransigent behaviors. Herein, the cytotoxicity effect of HA from *H. scabra* significantly inhibited the viability of PC3 cells at 24 and 48 h, with an IC_50_ of 1.68 and 1.56 µM, respectively. Similar to our findings, the cytotoxicity of HA was observed in a variety of cancer cell lines [39,40,41]. Following that, we further clarified an inhibitory effect of HA on the inhibition of the EMT and metastasis in metastatic phenotype of human PCa cell lines (PC3 and LNCaP) as well as overexpressed RUNX1 in PC3 cells through an underlying mechanism. We found that the HA treatment could antagonize the EMT program in endogenous and exogenous RUNX1-expressing PCa cells in a concentration-dependent manner. Especially, an HA treatment at 1 µM obviously exhibited the effective dosage in these PCa cells. Furthermore, our results demonstrated that HA downregulated the expression of the phosphorylated Akt/JNK and P38 MAPK pathway and was followed by a decreasing ectopic and endogenous RUNX1 expression in PCa cell lines. As a consequence of the impairment of the RUNX1 translocation, the process of EMT was inhibited by the upregulating expression of E-cadherin, and the downregulating expression of vimentin, twist1, slug, and snail, thus preventing the metastasis of PCa cell lines via decreasing MMP2 and MMP9 protein expressions. Consistent with this, other reports have revealed that triterpenoid saponins from plants are effective at inhibiting the invasion and metastasis of several types of cancer cells. For example, ginsenoside Rd has been shown to inhibit the invasion and migration of human hepatocellular carcinoma (HepG2) cells by inhibiting P38 MAPK phosphorylation and inducing a focal adhesion formation [42,43]. In a lung cancer cell line (A549), ginsenoside 20-Rg3 ameliorated the migration, invasion, and resistance to anoikis by inhibiting TGF-β-induced EMT [44]. Considering sea cucumbers, prior studies demonstrated that the triterpene glycoside frondoside A from *Cucumaria frondosa* could inhibit the migration of a human bladder cancer cell line (UM-UC-3) [45] and the MDA-MB-231 breast cancer cell line via the downregulation of the P38/MAPK and PI3K/Akt pathways [45]. The suppression of these signaling pathways also decreased MMP2 and MMP9 expressions, which are the critical markers of tumor metastasis [45]. Moreover, Hep G2 cell motility was ameliorated by 24-dehydroechinoside A (DHEA) from *Pearsonothuria graeffei* via reducing MMP9 expression as well as enhancing TIMP-1 expression as a result of inhibiting NF-κB activation [45]. However, we currently only know that saponins can break the phospholipid bilayer of the plasma membrane, thus entering the cancer cells for various targeted signaling molecules. Thus, further in-depth studies of molecular docking and in vivo studies are required to prove the interaction of HA with the targeted molecule and the approach of metastatic prostate cancer treatment, respectively. 

## 4. Materials and Methods

### 4.1. Cell Cultures and Reagents

Human PCa cell lines, PC3 (ATCC CRL-1435) and LNCaP (ATCC CRL-1740), were purchased from the American Type Culture Collections (ATCC, Manassas, VA, USA). The cells were cultured in Rosawell Park Memorial Institute (RPMI) 1640 medium, supplemented with 10% fetal bovine serum (FBS) and 1% penicillin/streptomycin, at 37 °C in a humidified atmosphere with 5% CO_2_. All media, sera, and culture reagents were purchased from Gibco (Carlsbad, CA, USA).

Standard HA used throughout this investigation was imported directly from our previous work [17] and was used as such with no further amendment. The integrity of the standard HA was examined by ^1^H NMR spectroscopy, from which no obvious adulterants were observed. In addition, docetaxel (Sigma-Aldrich, St. Louis, MO, USA), a standard chemotherapy drug for PCa, was used as a positive control. The handling and extraction protocol of *H. scabra* were approved by the Faculty of Science, Mahidol University Animal Care and Use Committee (MUSC-ACUC), permit no. MUSC63-002-510.

### 4.2. MTT Assay

Colorimetric assays were performed using an MTT assay. Briefly, 8 × 10^3^ PC3 and LNCaP cells were plated and cultured in media containing HA (0.3125 to 20 µM) for 24 and 48 h in a 37 °C incubator with 5% CO_2_. The same condition as for the MTT assay was applied for docetaxel (0.001 to 5 µM). Untreated cells were used as controls. At a completed time of incubation, the medium was removed and a 5 mg/mL MTT solution (Sigma-Aldrich, St. Louis, MO, USA) was added to each well and further incubated for 2 h at 37 °C. After the supernatant was carefully removed, the formazan crystals in each well were dissolved in 100 µL of DMSO. The absorbance was measured at 562 nm in a VersaMax microplate reader (Molecular Devices, San Jose, CA, USA). All the measurements were done in triplicate. The cell viability is presented as the percentage of control. 

### 4.3. Observation of Cell Morphology

PC3 or LNCaP cells were plated at 3 × 10^5^ cells in 60 mm dishes. At 70–80% confluence, cells were treated with 0.25, 0.5, and 1 µM HA, or 1 µM docetaxel for 24 h. Cell morphology was observed and photographed with an inverted ECLIPSE E200 upright microscope (Nikon Corporation, Minato-ku, Tokyo, Japan).

### 4.4. Expression Plasmids and Host Selection

Human runt-related transcription 1 (RUNX1) with Myc-DDK-tagged expression plasmids (OriGene Technologies, Inc., Rockville, MD, USA) were inserted into *Escherichia coli* (*E. coli*) strain DH5α, which provided the host for the plasmid transformation. Isopropyl-β-D-thiogalactopyranoside and 5-bromo-4-chloro-3-indolyl-β-D-galactoside (X-Gal) (Merck/ MilliporeSigma, Burlington, MA, USA) were used for the blue-white colony screening. A single *E. coli* colony was picked up to ensure the expression of the plasmid carrying RUNX1 by a QIAGEN plasmid minikit (QIAGEN, Germantown, MD, USA). 

### 4.5. Plasmid Extraction

Plasmids containing the RUNX1 gene were extracted using a plasmid minikit (Qiagen, Germantown, MD, USA) according to the manufacturer’s instructions. Briefly, after growing *E. coli* overnight, bacteria were collected by centrifugation at 4 °C, 6000× *g* for 15 min. Subsequently, the supernatant was discarded and buffer P1 was added to the pellet. Next, buffer P2 for cell lysis was added, mixed carefully by inverting 4–6 times, and incubated at room temperature for 5 min. Finally, buffer P3 was added to the mixture, and incubated on ice for 5 min followed by centrifugation at 18,000× *g*, for 10 min. Then, the supernatant containing plasmid DNA was added to the QIAGEN-tip and allowed it to penetrate the resin by gravity flow. Following the washing with buffer QC and centrifugation, DNA was eluted by adding buffer QF. The mixture was centrifuged at 4 °C, 18,000× *g*, for 30 min. After washing with 70% ethanol and drying at room temperature, the pellet was dissolved in RNase-free water. Finally, the DNA concentration was determined by NanoDrop 2000 spectrophotometers (Thermo Fisher Scientific, Waltham, MA, USA).

### 4.6. Cell Transfection

PC3 cells were seeded into sterile 60 mm cell culture dishes and incubated at 37 °C overnight to allow cell attachment. A lipofectamine 3000 reagent (Invitrogen, Carlsbad, CA, USA) was used to transfect plasmid DNA into PC3 cells. Following the manufacturer’s protocol, the lipofectamine reagent was mixed with 125 µL of Opti-MEM reduced serum medium (Gibco, Carlsbad, CA, USA). Next, 5 μg of plasmid DNA was diluted in 250 μL of Opti-MEM medium to prepare a master mix of DNA, then a P3000 reagent was added to a final concentration of 2 μL/μg DNA and well mixed. At the end, the diluted plasmid DNA was added to each tube of diluted Lipofectamine 3000 reagent at a 1:1 ratio (*v*/*v*) and incubated for 10–15 min at room temperature. The DNA–lipid complex was added to cells in the culture dish and incubated at 37 °C overnight. The efficiency of the cell transfection was verified by a Western blot analysis. 

### 4.7. Gene Expression Analysis by Quantitative RT-PCR

Total RNA was extracted using a total RNA minikit. Concisely, one microgram of RNA was treated with DNase I, and the first-strand cDNA was then synthesized using iScript cDNA synthesis kit. The real-time PCR was performed using iTaq Universal SYBR Green Supermix on the CFX96 Touch Real-Time PCR System and the C1000 Touch Thermal Cycler (Bio-Rad Laboratories, Hercules, CA, USA). Fold changes were calculated using the 2^−∆∆Ct^ method [46]. The relative fold change value of untreated cells was defined as 1. All kits and reagents for the qRT-PCR were purchased from Bio-Rad Laboratories (Hercules, CA, USA). The specific forward and reverse primers are listed in Appendix A.

### 4.8. Cell Migration and Invasion Assays

The transwell of a 24-well plate inserted with modified 8 µM pore size membrane chambers (BD Biosciences, Franklin Lakes, NJ, USA) was applied, and the invasive ability was tested using Matrigel-coated Biocoat cell culture inserts (BD Biosciences, Franklin Lakes, NJ, USA) with 8 µM pores in 24-well plates. A total of 1 × 10^5^ cells were plated in the upper compartment while the lower compartment was filled with 500 µL of RPMI growth medium containing 10% FBS, acting as a chemoattractant. Various concentrations of HA (0.25, 0.5, and 1 µM) were added to the upper portions of the chambers. Following 24 h of culture at 37 °C, cells were fixed with 100% ice-cold methanol for 15 min and washed twice with cold PBS. Afterwards, 1% crystal violet was added and incubated at room temperature for 20 min. Cells that failed to pass through the membrane between the upper and lower chambers were scraped. Cells that migrated or invaded through the inserts were photographed and counted at a magnification of 10× by a Nikon inverted microscope in five randomly selected fields. Five randomly selected fields at 10× magnification were counted. 

### 4.9. Western Blotting

Whole cellular proteins were extracted using a RIPA cell lysis buffer (Cell Signaling Technology, Danvers, MA, USA). The protein concentration was determined using a BCA assay kit (Thermo Fisher Scientific, Waltham, MA, USA). After 30 µg of total protein was loaded into the wells and separated by 12.5% SDS-PAGE, it was electrophoretically transferred to a nitrocellulose membrane. The membrane was blocked in Tris-buffered saline with Tween-20 (TBS-T) containing 5% BSA for 2 h and incubated overnight at 4 °C with primary antibodies as follows: Akt (dilution 1:1000; cat. no. 9272), phosphor-Akt (dilution 1:1000; cat. no. 4060), ERK (dilution 1:1000; cat. no. 4695), phospho-ERK (dilution 1:1000; cat. no. 4370), P38 (dilution 1:1000; cat. no. 8690), phospho-P38 (dilution 1:1000; cat. no. 4511), SAPK/JNK (dilution 1:1000; cat. no. 9252), phospho-SAPK/JNK (dilution 1:1000; cat. no. 4668), RUNX1 (dilution 1:700; cat. no. 4334), MMP2 (dilution 1:700; cat. no. 4022), MMP9 (dilution 1:700; cat. no. 13667), vimentin (dilution 1:1000; cat. no. 5741), twist1 (dilution 1:1000; cat. no. 69366), slug (dilution 1:1000; cat. no. 9585), snail (dilution 1:1000; cat. no. 3879), E-cadherin (dilution 1:1000; cat. no. 3195), and β-actin (dilution 1:1000; cat. no. 4970) served as a loading control; the antibodies were purchased from Cell Signaling Technology. Horseradish peroxidase-conjugated secondary antibodies (dilution 1:5000; cat. no. ab205718) (Abcam, Cambridge, MA, USA) were employed. Immunoreactive bands were detected by an enhanced chemiluminescence (ECL) technique using Pierce™ ECL Plus Western Blotting Substrate (Thermo Fisher Scientific, Waltham, MA, USA), as previously described [46].

### 4.10. Gelatin Zymography

To detect the gelatinase MMP 9 activity, gelatin zymography was carried out according to the method previously described by Sangpairoj et al. [7]. 

### 4.11. Statistical Analysis

All in vitro studies were performed in triplicate and results are expressed as mean ± S.D. Statistical significance was determined by means of a one-way ANOVA followed by Dunnett’s multiple comparisons test using GraphPad Prism version 8.0.0 for Windows (GraphPad Software, San Diego, CA, USA). A probability value of *p* < 0.05 was considered statistically significant in all calculations. 

## 5. Conclusions

Taken together, our findings established new insights into the use of a potent holothurin A as a PCa remedy. Not only did this approach suppress the prostate cancer EMT and metastasis through the regulation of the Akt/JNK and P38 MAPK pathways, but it also halted RUNX1-targeted EMT/metastasis strategies. This in turn could lead to the development of an effective prognostic RUNX1 factor and targeted therapy for advanced-stage prostate cancer, which requires further exploring in mammalian models.

## Figures and Tables

**Figure 1 marinedrugs-21-00345-f001:**
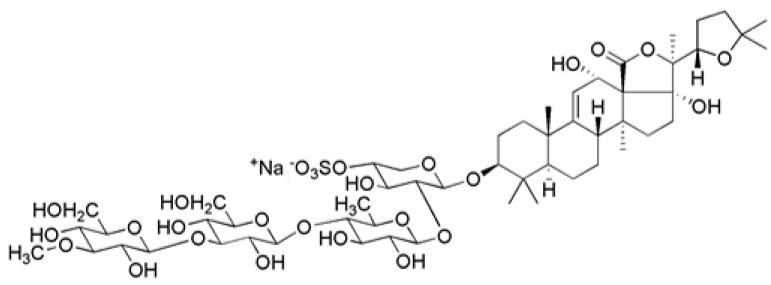
Chemical structure of holothurin A.

**Figure 2 marinedrugs-21-00345-f002:**
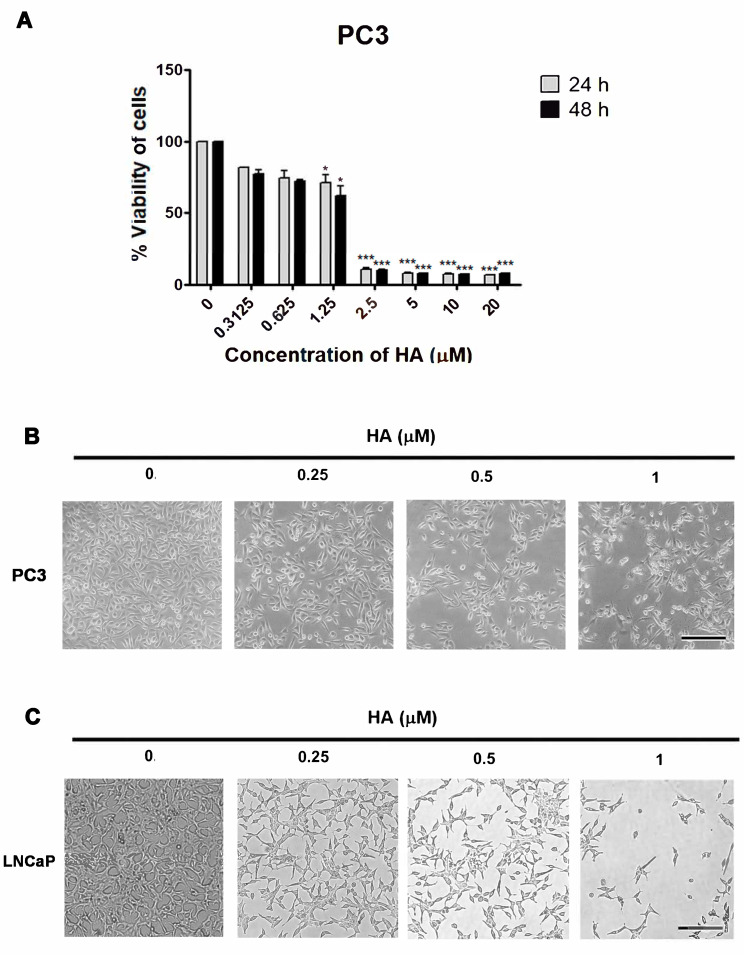
The effect of HA on the viability of human PCa cell lines; PC3 and LNCaP. (**A**) The results of the MTT assay showing the viability of PC3 prostate cancer cells after HA treatment. Bright field photographs show morphological changes in PC3 (**B**) and LNCaP (**C**) cells treated with either 0.25, 0.5, and 1 µM of HA at 24 h. HA, holothurin A. Scale bar, 100 µM. Values represent mean + SD from triplicates. * *p* < 0.05; *** *p* < 0.001.

**Figure 3 marinedrugs-21-00345-f003:**
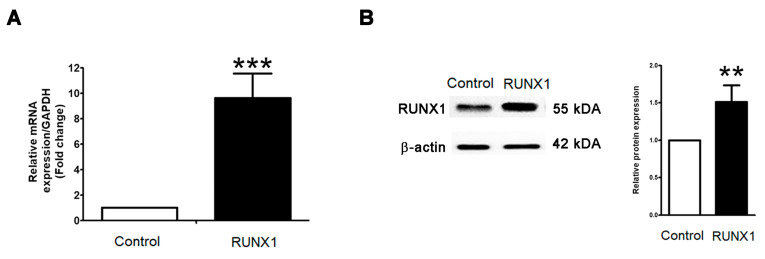
Overexpression of RUNX1 in PC3 cell line. RUNX1 plasmids were transfected in human prostate cancer PC3 cell lines, which were observed by mRNA and protein expressions. The relative mRNA and protein expression levels of RUNX1 after transfection were detected by qRT-PCR (**A**) and Western blot (**B**). ** *p* < 0.01; *** *p* < 0.001, compared with the vehicle control.

**Figure 4 marinedrugs-21-00345-f004:**
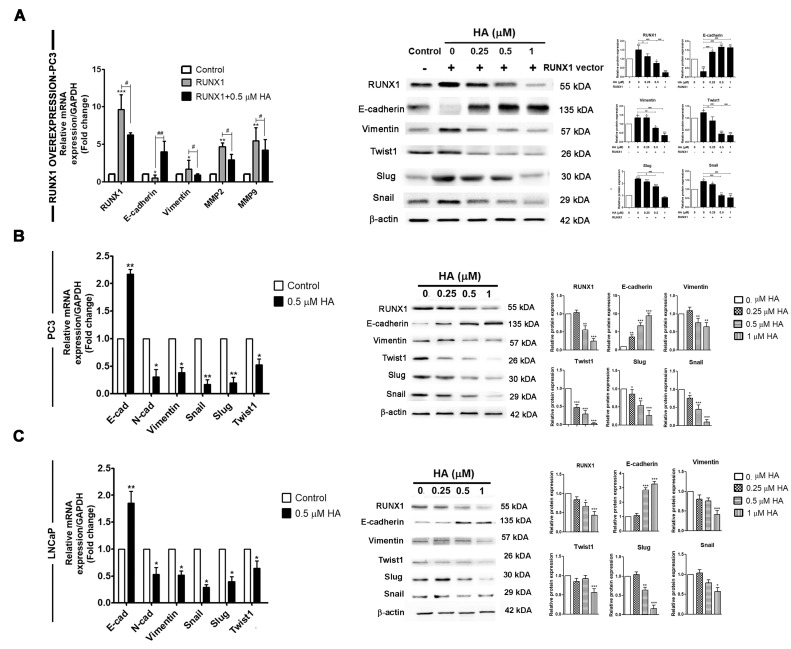
HA suppresses mRNA and protein expressions of EMT-related proteins of PCa cell lines. (**A**) Following RUNX1 transfection, HA at 0.25, 0.5, and 1 µM were treated for 24 h. Protein expression levels and qRT-PCR of EMT markers were examined in RUNX1-overexpressing PC3 prostate cancer cells. qRT-PCR and Western blotting were observed in HA-treated PC3 (**B**) and LNCaP (**C**) cells. Values are expressed as mean ± SD. HA, holothurin A. * *p* < 0.05; ** *p* < 0.01; *** *p* < 0.001, compared with the control. # *p* < 0.05; ## *p* < 0.01; ### *p* < 0.001, compared with the RUNX1-transfected group.

**Figure 5 marinedrugs-21-00345-f005:**
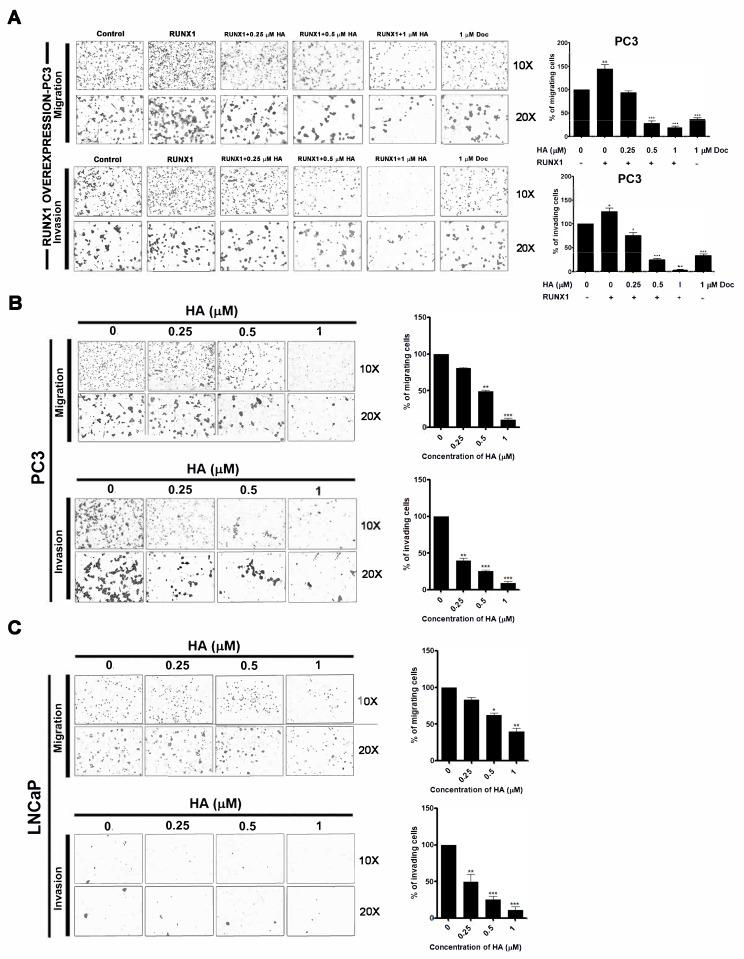
HA suppresses RUNX1-mediated migration and invasion of PCa cell lines. Transwell assays were performed to assess cell migration and invasion in RUNX1-overexpressing PC3 cells (**A**) followed by a 0.25, 0.5, and 1 µM HA treatment and HA treatment alone in PC3 (**B**) and LNCaP (**C**) cells. Doc is used as a positive control. Magnification, ×10 and ×20. Values are expressed as the mean + SD. HA, holothurin A; Doc, docetaxel. * *p* < 0.05; ** *p* < 0.01; *** *p* < 0.001, compared with the vehicle control.

**Figure 6 marinedrugs-21-00345-f006:**
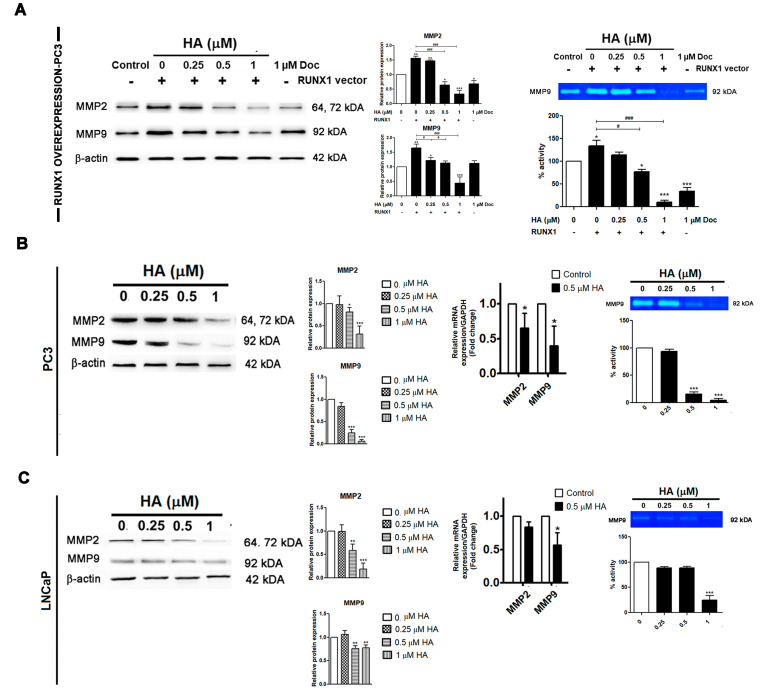
HA suppresses the RUNX 1-mediated enzymatic activity of MMPs of PCa cell lines. MMPs gelatinolytic activities were evaluated by gelatin zymography in RUNX1-overexpressing PC3 cells (**A**) followed by 0.25, 0.5, and 1 µM HA treatment and HA treatment alone in PC3 (**B**) and LNCaP (**C**) PCa cells. Doc is used as a positive control. HA, holothurin A; MMPs, matrix metalloproteinases; Doc, docetaxel. * *p* < 0.05; ** *p* < 0.01; *** *p* < 0.001, compared with the control. # *p* < 0.05; ### *p* < 0.001, compared with the RUNX1-transfected group.

**Figure 7 marinedrugs-21-00345-f007:**
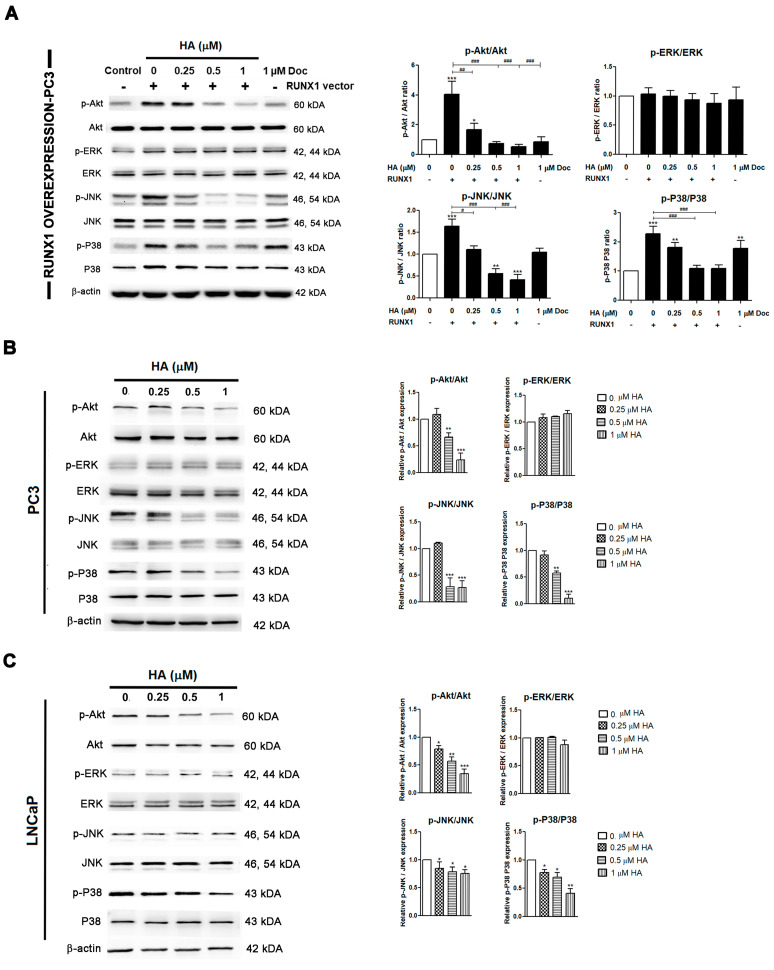
HA inhibits EMT-mediated metastasis via inhibiting Akt, JNK, and P38 MAPK signaling pathways. PC3 cells were transfected with RUNX1 followed by HA treatment at 0.25, 0.5, and 1 µM of docetaxel for 24 h. Protein expression levels of important genes in the MAPK pathway were examined by a Western blot analysis (**A**). HA treatment alone is observed in PC3 (**B**) and LNCaP (**C**) PCa cells. Doc is used as a positive control. HA, holothurin A; Doc, docetaxel. * *p* < 0.05; ** *p* < 0.01; *** *p* < 0.001, compared with the control. # *p* < 0.05; ## *p* < 0.01; ### *p* < 0.001, compared with the RUNX1-transfected group.

## Data Availability

The data supporting the conclusion in this study are available upon request from the corresponding author.

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
