# Peer review of "Holothurin A Inhibits RUNX1-Enhanced EMT in Metastasis Prostate Cancer via the Akt/JNK and P38 MAPK Signaling Pathway"

_marinedrugs, 2023, doi:10.3390/md21060345_

Round 1
Reviewer 1 Report
In this manuscript, the authors have determined the effect of Holothurin A on EMT-mediated metastasis of PCa cell lines, which with endogenous and exogenous RUNX1 expressions, via the assay of cell viability, cell migration and invasion, cell transfection etc.
Overall, the manuscript is well written- the rationale is well explained and the hard work of the authors were evident by the abundance of the results.
One suggestion:
Authors have examined the expression of RUNX1, E-cadherin, Akt ERK etc. on the levels of mRNA and protein respectively in order to figure out the effect of HA on EMT-mediated metastasis of PCa cell. As the interaction among genes is quite complicated, especially on how to active or block a signal pathway via up/down-regulation of mRNA and protein level. Thus, it could be helpful to add a figure which based on gene interaction network and illustrating how HA affect the signal pathways by differential expression.
Author Response
Reviewer's suggestion: Authors have examined the expression of RUNX1, E-cadherin, Akt ERK etc. on the levels of mRNA and protein respectively in order to figure out the effect of HA on EMT-mediated metastasis of PCa cell. As the interaction among genes is quite complicated, especially on how to active or block a signal pathway via up/down-regulation of mRNA and protein level. Thus, it could be helpful to add a figure which based on gene interaction network and illustrating how HA affect the signal pathways by differential expression.
Authors' response: Schematic representation of the mechanism of action based on gene interaction and how HA affect the signaling pathways is illustrated in graphical abstract as shown below.

Reviewer 2 Report
The manuscript describes the mechanism of Holothurin A, a triterpenoid saponin isolated from Holothuria scabra, in epithelial-mesenchymal transition (ETM)-driven metastasis of human PCa cell lines. The experiments were well-designed, results are interesting and well-written. I would like to recommend the manuscript for publication in Marine Drugs.
Author Response
Thank you so much for taking the time to give us a review. We would like to express our sincere gratitude for the time you took to review our manuscript. Your feedback was incredibly valuable to us.
Reviewer 3 Report
In this manuscript, the authors described that RUNX1 overexpression could promote the EMT phenotype with increased EMT markers, consequently driving metastatic migration and invasion in PC3 cell line through the activation of Akt/MAPK signaling pathways. Intriguingly, HA treatment could antagonize EMT program in endogenous and exogenous RUNX1 expressing PCa cell lines. Along with decreasing metastasis of HA-treated both cell lines were evidenced through down-regulation of MMP2 and MMP9 via Akt/P38/JNK-MAPK signaling pathway. Different efficient methods have been used to confirm the results and the results were clear for conclusion. The writing is logical and comprehensive. However, I do have some minor comments as follows:
1. In method of plasmid extraction, you should italic the Escherichia coli (E. coli).
2. In Figure 4A, you showed the effect of HA on the RUNX1 overexpression PC3. I suggest the author should add mRNA result of Snail, Slug and Twist.
3. EMT acts as a key of cancer metastasis, the authors should also mention the difference between RUNX1 overexpression PC3 with PC3 after treatment with HA in the discussion.
4. In Figure 6B, the bands of MMP2 and MMP9 in PC3 cells seem much thicker than those of RUNX-1 overexpression PC3 which enhanced of MMP2 and MMP9. The authors should carefully check this again.
Author Response
- Inmethod of plasmid extraction, you should italic the Escherichia coli (E. coli).
Response: Thank you so much for taking the time to give us a review. The Escherichia coli (E.coli) format is changed according to your suggestion on page 12, section 4.4 Expression plasmids and host selection.
- In Figure 4A, you showed the effect of HA on the RUNX1 overexpression PC3. I suggest the author should add mRNA result of Snail, Slug and Twist.
Response: We also agree with your valuable suggestion that mRNA result of Snail, Slug and Twist should be added into the manuscript. However, in this study, we used E-cadherin as the representative of the epithelial marker whereas vimentin is of mesenchymal marker. The results of both markers were shown in Figure 4A, left panel. Moreover, Instead of showing the mRNA expression of Snail, Slug and Twist, the protein expression of Snail, Slug and Twist were examined by Western blot analysis as shown in Figure 4A, right panel. In addition, due to a limited funding supported to our research, we cannot do all laboratory experiments suggested therefore we have chosen to do the Western blot analysis rather that the mRNA sequencing technique as it is cheaper. Furthermore, we have planned to do the mRNA examination in our future study.
- EMT acts as a key of cancer metastasis, the authors should also mention the difference between RUNX1 overexpression PC3 with PC3 after treatment with HA in the discussion.
Response: We thank to the reviewer’ s suggestion and add more details on page 11 in the discussion section “We found that HA treatment could antagonize the EMT program in endogenous and exogenous RUNX1 expressing PCa cells in a concentration-dependent manner. Especially, HA treatment at 1 µM obviously exhibited the effective dosage in these PCa cells.”
- In Figure 6B, the bands of MMP2 and MMP9 in PC3 cells seem much thicker than those of RUNX-1 overexpression PC3 which enhanced of MMP2 and MMP9. The authors should carefully check this again.
Response: Thank you so much for your invaluable advice. We carefully check and confirm this data. Western blot of MMP2 and MMP9 in PC3 cell line are thicker than those of RUNX1-overeexpression PC3 may result from the effect of passage number of the cell used. It is generally accepted that cell lines with higher passage numbers resulting in alteration of cell morphology and functions (Anderle et al., 1998). In this study, PC3 cells treated with HA was in passage between 20-22 whereas for RUNX1-overexpression PC3 was in passage between 20-30. However, passage lower than 30 is considered as young passages and is appropriate for an in vitro study (Lin et al.2003).
References:
- Anderle P, Niederer E, Rubas W, Hilgendorf C, Spahn-Langguth H, Wunderli-Allenspach H, Merkle HP, Langguth P. P-glycoprotein (P-gp) mediated efflux in Caco-2 cell monolayers:The influence of culturing conditions and drug exposure on P-gp expression levels. Journal of pharmaceutical sciences. 1998;87(6):757–762.
- Lin H et al. Suppression Versus Induction of Androgen Receptor Functions by the Phosphatidylinositol 3-Kinase/Akt Pathway in Prostate Cancer LNCaP Cells with Different Passage Numbers. Journal of Biological Chemistry. 51: 50902-50907, 2003.